# Numerical Simulation on Laser Shock Peening of B_4_C-TiB_2_ Composite Ceramics

**DOI:** 10.3390/ma16031033

**Published:** 2023-01-23

**Authors:** Xin Wang, Beidi Chen, Fan Zhang, Lisheng Liu, Shuang Xu, Hai Mei, Xin Lai, Lin Ren

**Affiliations:** 1School of Science, Wuhan University of Technology, Wuhan 430070, China; 2School of Materials Science and Engineering, Wuhan University of Technology, Wuhan 430070, China

**Keywords:** laser shock peening, numerical simulation, extended Drucker–Prager model, B_4_C-TiB_2_ ceramics

## Abstract

The introduction of residual stresses using laser shock peening (LSP) is an effective means of improving the mechanical properties of ceramics. Numerical simulations offer greater convenience and efficiency than in-lab experiments when testing the effects of different processing techniques on residual stress distribution. In this work, a B_4_C-TiB_2_ ceramic model based on the extended Drucker–Prager model was established to investigate the effects of laser power density, the number of impacts and laser spot overlapping rate on the residual stress distribution, and the reliability of the simulation method was verified by experimental data. The following results are obtained: increasing the laser power density and the number of impacts can increase the surface residual compressive stress and reduce the depth of the residual compressive stress; the presence of multiple impacts will significantly reduce the depth of the residual compressive stress layer; with the increase in the laser spot overlapping rate, the compressive residual stress in the processed area gradually increases and is more uniformly distributed; the best processing effect can be achieved by using a spot overlapping rate of 50%.

## 1. Introduction

B_4_C-TiB_2_, a composite ceramic material, has high hardness, low density, high toughness, and high electrical conductivity, making it a very promising complex ceramic material that has received a great deal of attention in recent years. Thus far, researchers have carried out a large number of studies on B_4_C-TiB_2_ ceramic. They have studied the influence of factors, such as relative density, grain size, and content of the toughening phase, on the toughness and hardness of B_4_C-TiB_2_ ceramics through different methods, such as pressureless sintering, hot pressing sintering, and spark plasma sintering, and achieved fruitful research results [1,2,3]. At the same time, however, it is difficult to further improve this material. The introduction of residual stresses by laser shock peening is one of the few effective means to improve the mechanical properties of ceramics.

Laser shock peening uses high-voltage plasma shock waves generated during the interaction between high-power short-pulse lasers and metal or ceramic materials so that the material has high residual compressive stress after the impact [4,5]. In comparison to general material modification methods (such as mechanical shot peening, cold extrusion, laser heat treatment, etc.), laser shock peening offers several unique advantages, such as its non-contact nature, efficiency, flexibility, lack of a heat-affected zone, and substantial strengthening effect, and it is widely used in aerospace, the nuclear industry, energy development, the automotive industry, machinery, and many other fields [6,7].

In 1972, Fairand et al. [8] from Battlell’s Columbus Laboratory in the United States first used high-power pulsed-laser-induced shock waves to change the microstructure and mechanical properties of 7075 aluminum alloy. The yield strength of the material was increased by 30% after shot peening, which opened the door to further research on the application of laser shot peening to various materials. In experimental studies, scholars have made great progress in improving metallic material properties such as fatigue life [9]; fracture toughness [10]; hardness and wear resistance [11]; strain rate [12]; residual stress [13]; material microstructure [14]. Currently, there are few studies on the application of LSP to ceramic materials. Wang et al. [15] used LSP to process polycrystalline SiC ceramics. The results show that a high residual compressive stress is generated on the surface of SiC ceramics during LSP, which can extend to a depth of 750 μm. The fracture toughness and bending strength of SiC ceramics were increased by 67% and 17%, respectively, after LSP treatment. Pratik Shukla et al. [16,17] investigated the fracture toughness and residual stresses on SiC and Al_2_O_3_ ceramic surfaces before and after LSP treatment. The results showed an increase in fracture toughness and residual compressive stress for both ceramics. The above study shows that LSP treatment can form a residual compressive stress layer on the surface of ceramics, improving the mechanical properties of the samples and increasing their fracture toughness and bending strength. Due to the high cost of experiments and the difficulty of observing the change process of the microstructure of materials during LSP, numerical simulation has become an effective tool to study LSP.

In 1999, Braisted et al. [18] used Abaqus finite element software to simulate the process of laser shock peening TI-6AL-4V titanium alloy and predicted the distribution of the residual stress field of 35CD4 stainless steel, which laid the foundation for the simulation of LSP by scholars afterward. After years of development, the finite element simulation of LSP is maturing [19,20,21]. The effect of key LSP parameters such as laser energy [22], laser spot overlapping rate [23], number of impacts [24], and laser spot shape [25] on the residual stress distribution was investigated using finite element simulations. Zhang et al. [26] carried out finite element simulations of double-sided laser shock peening on a thin plate of 7075-T7351 alloy to study the propagation and decay of LSP-induced shock waves in the plate thickness direction. The simulation results showed that the stress wave energy gradually decayed with the increase in propagation distance. Xiang et al. [27] conducted double-sided laser shock peening of 2024-T351 alloy blade by finite element simulation and experiment, and investigated the mechanism of the effect of laser plasma-induced shock wave reflection coupling on LSP-induced residual stress distribution, and the results showed that alternating double-sided LSP could produce better residual stress distribution. Most of these finite element simulations use a two-step analysis method of “dynamic explicit analysis + static implicit analysis” [28], which requires a lot of computational resources and is cumbersome when analyzing large models and multi-point impact problems. Scholars have proposed a continuous dynamic display analysis method [29,30]. This method uses CIN3D8 infinite elements as reflection-free boundaries to build a local model, which avoids the effects caused by the repeated transmission of stress waves inside the material and shortens the computational time. They have achieved fruitful research results in the field of metal materials by using the Johnson–Cook constitutive model; however, the finite element analysis method of laser shock peening ceramic materials has been less studied.

In this paper, the extended Drucker–Prager (D-P) constitutive model and continuous dynamic impact strategy were used to simulate the finite element analysis of applying LSP to B_4_C-TiB_2_ ceramics. The solution time of a single impact is determined, and the feasibility of the simulation method is verified by experiments. The effects of power density, impact times, and spot bonding ratio on the residual stress distribution were analyzed. This work provides the theoretical basis and a data reference for the process design and engineering applications of the laser impact strengthening of B_4_C-TiB_2_ ceramics.

## 2. Finite Element Simulation

### 2.1. Establishment of the Finite Element Model

The simulation object of this paper is hot pressing sintered B_4_C-TiB_2_ ceramics (heated to 2050 °C under 30 MPa pressure and held for 1 h). The mass fractions of B_4_C and TiB_2_ are 80% and 20%, respectively. The mechanical properties of B_4_C and TiB_2_ are shown in Table 1, where ρ is the material density,E is the elastic modulus,ν is the Poisson’s ratio, and G is the shear modulus.

Ceramics will undergo dynamic plastic deformation with an ultra-high strain rate under the action of the laser shock wave. Of the existing constitutive equations describing brittle materials, the extended Drucker–Prager model can be used in tandem with the Johnson–Cook rate correlation model to comprehensively consider the response of ceramic materials under large strain, high strain rate, and high-pressure impact conditions. In this paper, we use a calibrated Drucker–Prager plasticity model and an equation of state to obtain a material behavior that is similar to that of the JH-2 model. In this way, we can no longer limit ourselves to specific expressions of the JH-2 model [31], as well as material parameters, and thus obtain better simulation results. The strength model uses the general exponent form of the extended D-P model, which can be represented as follows:(1)$$\sigma =\frac{1}{a^{1/b}}{\left(P+P_{t}\right)}^{1/b}$$
where σ is the von Mises equivalent stress;P is the actual pressure; $P_{t}$ is the hardening parameter; a and b are constants.

We find that the strength expression of the JH-2 model can also be deformed into a similar expression, as in Equation (2)
(2)$$\sigma_{i}=\frac{\sigma_{HEL}A}{P_{HEL}^{N}}{\left(P+T\right)}^{N}$$
where $P_{HEL}$ is the pressure at the HEL; $\sigma_{HEL}$ is the equivalent stress at the HEL; A and N are the material constants under the JH-2 model. T is the maximum tensile pressure that the material can withstand.

Comparing these two expressions, the equations to calibrate the material parameters in the Drucker–Prager model can be obtained as follows:(3)$$a=\frac{P_{HEL}}{{\left(A\sigma_{HEL}\right)}^{1/N}}$$
(4)b=1/N
(5)$$P_{t}=T$$

The uniaxial compression yield stress $\sigma_{c}$ can be found by the following equation:(6)$$P_{t}=a\sigma_{c}^{b}-\sigma_{c}/3$$

The effect from strain rates is incorporated by the Johnson–Cook strain rate dependence law as follows:(7)$$\sigma =\sigma_{0}(1+C\ln\frac{\dot{\varepsilon }}{{\dot{\varepsilon }}_{0}})$$
where $\dot{\varepsilon }$ is the strain rate; ${\dot{\varepsilon }}_{0}$ is the reference strain rate; C is the strain enhancement factor.

The hydrodynamic behavior of ceramic materials is described by the Mie–Grüneisen equation of state. The linear $U_{s}-U_{p}$ Hugoniot form is used. The pressure is expressed as follows:(8)$$P=\frac{\rho_{0}c_{0}^{2}}{{\left(1-s\eta \right)}^{2}}$$
where $\eta =1-\rho_{0}/\rho =\mu /(1+\mu )$ is the nominal volumetric compressive strain; $c_{0}$ is the bulk speed of sound; s is the slope of the linear $U_{s}-U_{p}$ Hugoniot form of the equation of state; $\rho_{0}$ is the reference density; ρ is the current density.

The pressure–density relationship equation for the JH-2 model is as follows:(9)$$P=K_{1}\mu +K_{2}\mu^{2}+K_{3}\mu^{3}$$
where $K_{1}$, $K_{2}$ and $K_{3}$ are constants. Combining the above two equations, we can solve for s and $c_{0}$. D-P model parameters of B_4_C-TiB_2_ as shown in Table 2.

In order to analyze the stress wave propagation process under a single laser impact and study the effects of the power density and the number of impacts on the residual stress distribution, a single-point impact model with a size of 4 mm × 4 mm × 3 mm was established, as shown in Figure 1a. To study the effect of the laser spot overlapping rate on the residual stress distribution, a 16-point impact model of 10 mm × 10 mm × 3 mm was established, as shown in Figure 1b. The laser spot diameter of both models is 2 mm, and the model body uses C3D8R elements with a mesh size of 0.03 mm. Except for the impacted surface, other surfaces are set as infinite boundaries using CIN3D8 elements to eliminate the effect of repeated transfer of stress waves within the model on the simulation results.

### 2.2. Determination of Impact Parameters

Figure 2a shows the schematic diagram of the laser impact peening of ceramics. The ceramic sample is covered by a sacrificial layer and a plasma-confining medium. Plasma is formed on the ceramic surface by laser irradiation, and the explosive expansion of the plasma generates shock waves that propagate into the ceramic material. Industrial black tape is generally used as a sacrificial layer, mainly to prevent laser ablation. The plasma-confining medium is deionized water.

French scholar R. Fabbro et al. [32] derived the estimation formula for laser shock wave peak pressure (GPa) through experimental measurement and theory, as shown in Equation (3) as follows:(10)$$P_{\max}=0.01\sqrt{\frac{\alpha }{2\alpha +3}}\sqrt{ZI}$$
where α is the efficiency coefficient of internal energy conversion into thermal energy, which was 0.1; I is the laser power density (GW/cm^2^); Z is the reduced shock impedance between the target and the plasma-confining water, and its calculation formula is Equation (4) as follows:(11)$$\frac{2}{Z}=\frac{1}{Z_{to}}+\frac{1}{Z_{co}}$$
where $Z_{to}$ and $Z_{co}$ are the shock impedance of the ceramic target and the plasma-confining water.

The spatial and temporal distribution of laser shock waves shows that the shock wave pressure follows the Gaussian distribution (Figure 2b) with radial distance within the laser spot, and the duration is about 2 –3 times that of the laser pulse [32]. According to the test results of the PVDF piezoelectric sensor, the shock wave pressure change curve with time can be simplified to a triangular distribution, as shown in Figure 2c. American scholar Zhang et al. [33] proposed that the distribution of pressure with the radial size and time of the spot during laser shock intensification is shown by Equation (5) as follows:(12)$$P(r,t)=P_{\max}P(t)\exp(-\frac{r^{2}}{2R_{0}^{2}})$$
where P(r,t) is the pressure at the distance r from the spot center at time t; P(t) is the pressure amplitude at the spot center at time t; $R_{0}$ is the spot diameter; r is the distance from the spot center.

After the laser shock wave acts on the surface of the target, several types of stress wave interactions are generated inside the material that takes a long time to stabilize. It has been shown in [34] that when the internal energy W_i_ of the material remains constant and the kinetic energy W_k_ of the material tends to zero, the residual stress field inside the material can be considered to be stabilized, and the optimal solution time for a single shock can also be obtained from this. The energy change curve of a single point impact is shown in Figure 3. When t = 060 ns, the internal energy W_i_ and kinetic energy W_k_ of the material increase rapidly under the action of the laser shock wave, reaching 14 mJ and 11 mJ, respectively, and then gradually decaying; when t > 1000 ns, the internal energy W_i_ stabilizes at 5 mJ and the kinetic energy W_k_ tends to 0 mJ. Thus, the solution time of a single impact can be determined as 1000 ns.

## 3. Simulation Results and Analysis

### 3.1. Analysis of stress WAVE Propagation Process

When a laser-induced high-pressure plasma shock wave acts on the surface of the sample, a stress wave is generated inside it and results in an ultra-high strain rate dynamic response from the specimen. Figure 4 shows the propagation process of the stress wave inside the specimen under a single-point impact, with the impact position at the center of the upper surface of the specimen and the power density chosen as 5.09 GW/cm^2^.

Figure 4a shows that the laser shock loading stage is at 90 ns, the laser-induced high-pressure plasma shock wave acts on the specimen, and the compressional stress wave and tensile unloading wave are generated successively and begin to propagate in the depth direction. At 220 ns, the compressional stress wave propagates to the lower surface, and the peaks of the compressional stress wave and tensile unloading wave decrease significantly (Figure 4b). At 380 ns, two stress waves propagate out of the infinite boundary (Figure 4c). Finally, at 1000 ns, the stress waves inside the specimen remain stable, and a stable residual stress field is formed on the surface and inside the specimen (Figure 4d).

As can be seen from Figure 4, the stress wave decayed rapidly during the propagation process. The maximum compressive stress value at the impact point surface decreases from 1375 MPa at 90 ns to 651 MPa at 1000 ns, which is analyzed to be due to the energy dissipation through heat generation, defects, and other means during the LSP process. Therefore, the initial compressive stress wave and tensile unloading wave generated during the laser impact loading phase have a large effect on the residual stress distribution.

### 3.2. Effect of Power Density on Residual Stress Distribution

Figure 5a shows that after a single-point LSP, the residual compressive stress is the largest at the center of the point of impact, gradually decreases along the radial direction, and becomes involved in tensile stress at the edge of the point of impact. This is because the shock wave pressure follows a Gaussian distribution along the radial distance in the laser spot. The residual tensile stress at the edge of the spot is due to the stress balance inside the material. With the increase in laser power density, the compressive residual stress on the ceramic surface gradually increases, and the maximum compressive residual stress increases from 411.33 MPa to 646.89 MPa. It is concluded that with the increase in power density, the impact load on the surface of the specimen and the surface plastic deformation increase, so the surface residual compressive stress also increases. The depth curve of residual stress (Figure 5b) shows that the residual compressive stress gradually decreases with the increase in depth, and the depth of the residual compressive stress layer decreases with the increase in power density. It is concluded that a compressive stress wave and a tensile unloading wave will be generated, transmitted, and gradually attenuated on the impact surface in the laser-impact loading stage. The compressive stress wave is first transmitted downward from the surface of the material, resulting in local plastic deformation of the ceramic material, and the residual compressive stress layer is then generated. The residual compressive stress decreases due to the attenuation of the compressive wave’s energy. The tensile unloading wave acts on the residual compressive stress layer and acts as plastic unloading. The greater the tensile wave’s energy is, the stronger the plastic unloading effect will be. Therefore, the depth of the residual compressive stress layer decreases with the increase in the power density.

### 3.3. Effect of the Number of Impacts on the Residual Stress Distribution

The number of times ceramic materials are allowed to strengthen at a fixed power density is one of the key parameters in LSP processing. Figure 6a shows that the surface residual stresses induced by LSP gradually increase with the increase in the number of shocks, and these stresses are −605.016 MPa, −877.095 MPa, and −1384.04 MPa, respectively. The increased amplitude increases with the increase in the number of shocks. The analysis shows that when the ceramic material is subjected to multiple impact loads, the damage will gradually accumulate, at which point the material enters the plastic softening stage, and the strength gradually decreases. Therefore, the increased amplitude of the residual compressive stress on the surface of the material will increase with the increase in the impact time. Figure 6b shows that the depth of the residual compressive stress layer decreases from 0.24 mm to 0.09 mm, and the decreased amplitude increases with the increase in the impact times. This is because the plastic unloading effect of the tensile unloading wave is stronger in the case of multiple impacts, which is not conducive to the stability of the high residual compressive stress on the surface.

### 3.4. Effect of Laser Spot Overlapping Rate on Residual Stress Distribution

Figure 7 shows the residual stress distribution under different spot overlapping rates. Here, the laser power density is 5.09 GW/cm^2^, and the spot diameter is 2 mm. When the spot overlapping rate is 0 (Figure 7a), the residual compressive stress in the central area of each spot is the largest, the residual compressive stress between the impact points is the smallest, and there is even residual tensile stress locally. This indicates that a lap ratio of zero can easily lead to a lap gap and forms residual tensile stress on the ceramic surface, resulting in greater stress concentration and stress mutation, which impacts the strengthening effect. When the spot overlapping rate is 25% (Figure 7b), the residual compressive stress gap is reduced, and a small amount of residual tensile stress is still generated in the impact domain. When the spot overlapping rate is 50% (Figure 7c), the residual compressive stress increases, and the distribution is more uniform, which indicates that the lapped gap is eliminated. When the spot overlapping rate is 75% (Figure 7d), the residual stress in the impact region further increases, but the residual stress in the region first strengthened by laser shock disappears due to the unloading wave. This indicates that the multi-point LSP with a 50% spot overlapping rate has the best effect, which can effectively avoid voids in the residual compressive stress bonding and plastic unloading.

### 3.5. Experimental Verification

To verify the feasibility and accuracy of the simulation method, single-point laser impact strengthening experiments were carried out to measure the residual stress distribution in the depth direction of the specimen and compare it with the simulation results. The experiments were carried out using a YD60-R200B laser impact intensifier developed by the Xi’an Tianruida company, with an output energy range of 2–6 J, a pulse width of 18–30 ns, an operating frequency of 1–5 Hz, a spot diameter of 2–6 mm, and a wavelength of 1064 nm. The residual stress measurement equipment is an LXRD-type residual stress tester from Proto Canada; the target material is Cu; the filter material is Ni; the diffraction angle 2θ is 135.5°; the diameter of the collimation tube is 2 mm; the number of beta angles is 9 and the maximum beta angle is 30°; the voltage is 35 KV; the current is 30mA; the exposure time is 2 s; the experiment is run 10 times. LSP experiments were conducted on B_4_C-TiB_2_ ceramics. The size of the sample is 20 × 20 × 8 mm. Before the experiment, the sample will be polished on both sides, black tape will be pasted on the upper surface, and the lower surface will be evenly coated with silicone grease to make the sample fit the experimental platform closely. Clamps were used to fix the sample on the experimental platform. The sample surface was covered with a layer of flowing plasma-confining water. The laser power density was set at 5.09 GW/cm^2^, the spot diameter was 2 mm, the lap rate was 50%, and the impact was performed 16 times. The residual stress was measured at seven points along the X-direction on the sample surface, and then the sample was polished layer by layer. The average residual stress was measured at 50 μm intervals.

Figure 8a shows the residual stress distribution along the X-direction on the sample surface, with the simulated average value of −195.79 MPa and the experimental average value of −240.43 MPa. Figure 8b shows that the average value of residual stress along the depth direction in the impact area calculated by simulation is reduced from −195.79 MPa to −19.236 MPa, and the depth of residual compressive stress is affected by 210 μm; the average value measured experimentally is reduced from −240.43 MPa to −11 MPa, and the depth of residual compressive stress is affected by about 250 μm. The simulation results do not exactly agree with the experimental results, and the reasons for the difference in results may be the following. The simulation model does not consider the effect of microstructures within the material, such as dislocations, vacancies, slip bands, twins, voids, inclusions, etc., on the LSP results. The slow decline of the residual stress with time after the LSP test may make the experimental results small. The layer-by-layer grinding and polishing process can affect the residual stress distribution on the sample surface. There is an evident agreement between the experimental and simulation results in terms of the variation trend, and the simulated values have a small error with the experimental results. Therefore, it is clear that the simulation method is feasible and has a high degree of accuracy.

## 4. Conclusions

In this paper, a B4C-TiB2 ceramic model based on the extended Drucker–Prager model was established to investigate the effects of laser power density, the number of impacts, and spot overlapping rate on the residual stress distribution, and the reliability of the simulation method was discussed through experimental data analysis. The following conclusions can be drawn from the results:The propagation process of the single-point impact stress waves within the sample was analyzed. The stress wave decayed rapidly during the propagation process, which made the initial compressive stress wave and tensile unloading wave have a large effect on the residual stress distribution;Increasing the laser power density and impact times can increase the surface residual compressive stress and reduce the depth of the residual compressive stress. In the case of multiple impacts, the plastic unloading effect of the tensile unloading wave is stronger, which can significantly reduce the depth of the residual compressive stress layer. Therefore, the strengthening effect of increasing the power density of a single impact is better;With the increase in the spot overlapping rate, the compressive residual stress in the impact domain increases gradually, and the distribution becomes more uniform. Using a 50% spot overlapping rate can effectively avoid residual compressive stress lap gaps and plastic unloading, in addition to the strengthening effect being the best at this ratio;The trends of the residual compressive stresses on the surface of the specimens obtained from the experiments and simulations were consistent, and the magnitudes of the values basically matched, indicating that the present constitution model and simulation method are feasible and have some reference significance.

## Figures and Tables

**Figure 1 materials-16-01033-f001:**
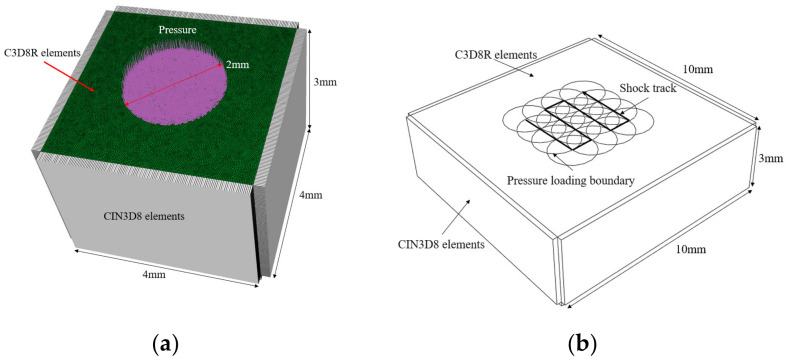
Schematic diagram of single-point impact (**a**) and multi-point impact (**b**) models.

**Figure 2 materials-16-01033-f002:**
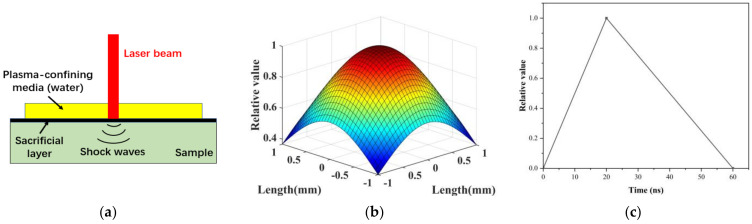
(**a**) Principle diagram of laser impact peening of ceramics. (**b**) The spatial distribution of shock wave pressure. (**c**) Time-dependent pressure distribution.

**Figure 3 materials-16-01033-f003:**
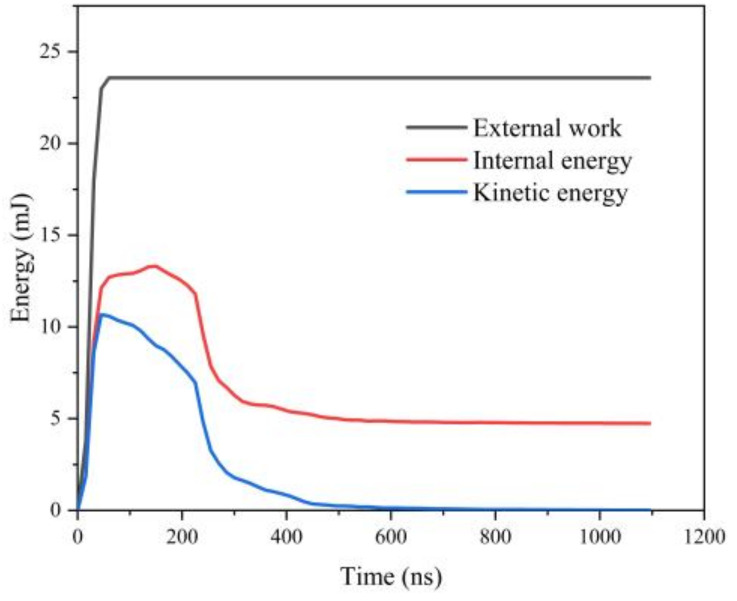
Variation curve of single-point impact energy.

**Figure 4 materials-16-01033-f004:**
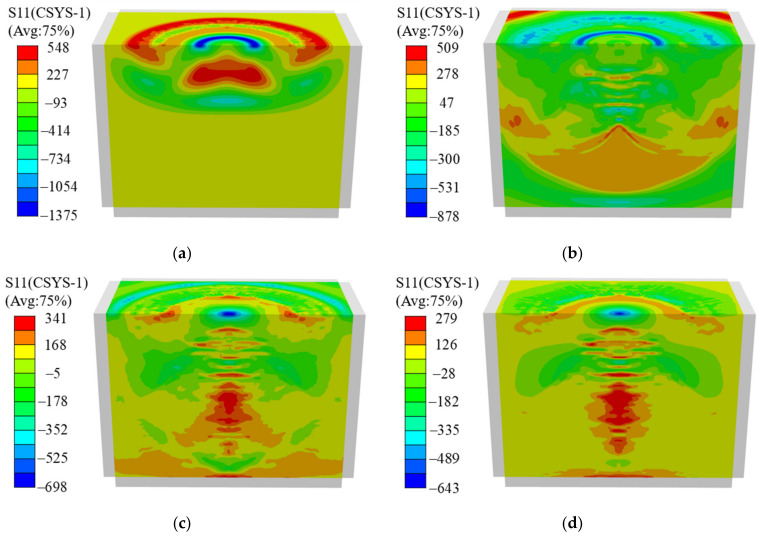
Single-point impact stress wave propagating in samples at different times of (**a**) 90 ns, (**b**) 220 ns, (**c**) 380 ns, and (**d**) 1000 ns.

**Figure 5 materials-16-01033-f005:**
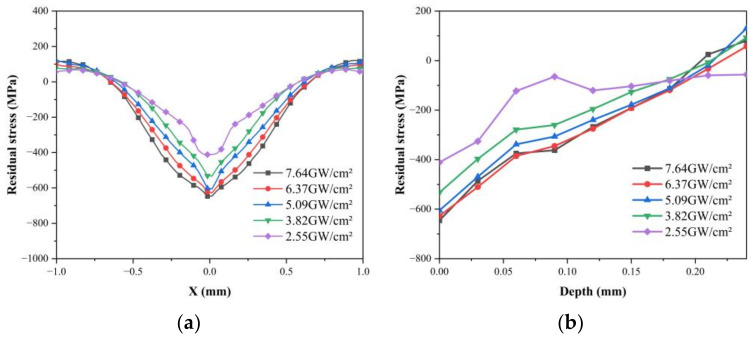
Under different power densities: (**a**) the residual stress distribution in the horizontal direction of the surface; (**b**) the residual stress distribution along the depth direction.

**Figure 6 materials-16-01033-f006:**
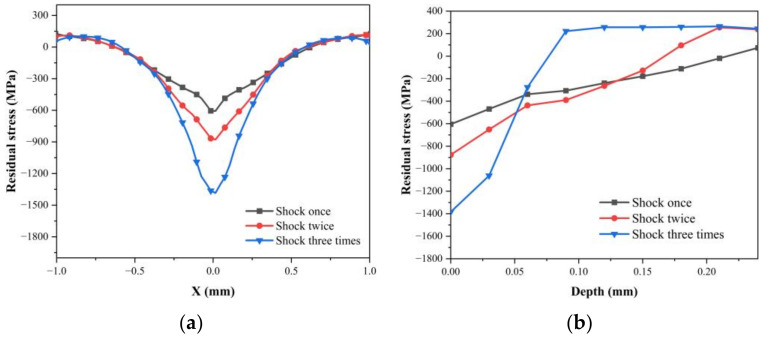
After multiple impact intensifications: the residual stress distribution in the horizontal direction of the surface (**a**); the residual stress distribution along the depth direction (**b**).

**Figure 7 materials-16-01033-f007:**
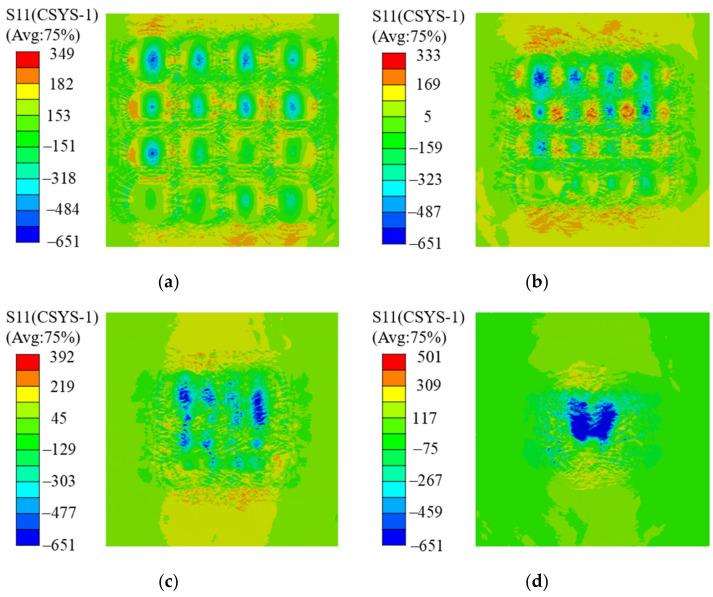
Distribution cloud diagram of residual stress at (**a**) 0%, (**b**) 25%, (**c**) 50%, and (**d**) 75% lap rates.

**Figure 8 materials-16-01033-f008:**
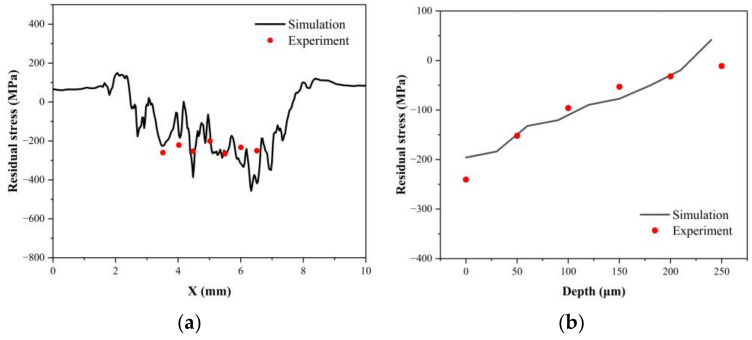
Comparison of experimental and simulated residual stress distribution (**a**) in the horizontal direction of the surface and (**b**) along the depth direction.

**Table 1 materials-16-01033-t001:** Mechanical properties and parameters of B_4_C-TiB_2_.

Mechanical Property	$\rho /\left(kg/m^{3}\right)$	E/GPa	ν	G/MPa
Value	2600	460	0.17	197

**Table 2 materials-16-01033-t002:** D-P model parameters of B_4_C-TiB_2_.

Parameters	a	b	$\sigma_{c}$	C	${\dot{\varepsilon }}_{0}$	s	$c_{0}$
Value	0.005483	1.492537	5483.4058	0.009	1.0	1.7725	9466.54

## Data Availability

Data sharing is not applicable to this article.

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
