# Peer review of "Numerical Simulation on Laser Shock Peening of B4C-TiB2 Composite Ceramics"

_materials, 2023, doi:10.3390/ma16031033_

Round 1

Reviewer 1 Report

The authors propose a numerical investigation of Laser Shock Peening-induced residual stresses on composite ceramic samples.

In general, the topic is of high interest, even if similar results can be found in literature. The paper is well written and clear, but many points need to be clarified and extended.

The abstract and the introduction are thorough in the definition of the motivation and goals of the paper, but the literature review should be extended mostly for the numerical simulations. Only two papers are cited as reference.

In lines 94-95, the discussion on Johnson–Holmquist-II (JH-2) constitutive model should be extended and detailed. Why the results of the simulations can be considered better?

It is not clear how the material properties are considering the hardening parameters, usually evaluated at high strain rates via a split-Hopkinson tension bar. The maximum strain rate measured in the tests is around 103 s-1, which is lower than that occurring during laser peening (i.e., in the order of 106 s-1) as stated by Peyre, P.; Fabbro, R.; Merrien, P.; Lieurade, H.P. Laser shock processing of aluminium alloys. Application to high cycle fatigue behaviour. Mater. Sci. Eng. A 1996, 210, 102–113.

However, must be checked if the ceramic material considered in the present paper has a low sensitivity on the strain rate. Moreover, a study by Langer et al. (Langer, K.; Olson, S.; Brockman, R.; Braisted, W.; Spradlin, T.; Fitzpatrick, M.E. High strain-rate material model validation for laser peening simulation. J. Eng. 2015, 13, 150–157) investigated the effects of using conventional test data to model the laser peening process; they concluded that consistent results could be obtained with strain rates in the order of 103 s-1.

The numerical models for the single shot (4x4x3mm) and overlap (10x10x3) are of different dimensions. To avoid any difference due to the mesh dimension and numerical boundary conditions, is recommended to have a numerical model with same size. Furthermore, there is a discrepancy in the thickness between the figure 1a dimensions and the text.

Are the numerical models representing therefore plates with 3 mm thickness? Or is just a small element of a thicker sample? This is a very important point because the thickness is a fundamental parameter in the shock wave propagation. The lower surface is reflecting the shock waves, creating therefore a Bauschinger effect in the material that is not taken into account in the simulations. The material response to the shock waves propagation can be thought of as a condition of high-strain rate cyclic loading. The effect of the cyclic deformation induced by laser peening can be simulated by a nonlinear isotropic/kinematic hardening plastic model, which also includes the change of properties due to the Bauschinger effect (see Angulo et al, The effect of material cyclic deformation properties on residual stress generation by laser shock processing, Int. J. Mech. Sci. 2019, 156; Troiani et al, The Effect of Laser Peening without Coating on the Fatigue of a 6082-T6 Aluminium Alloy with a Curved Notch, Metals, 2019)

The stress distribution induced by the LSP treatment is therefore totally different in thin and thick samples, as shown by some authors (check Troiani et al, Fatigue crack growth in laser shock peened thin metallic panels, Advanced Materials Research, 2014, 996; or Taddia et al, Effect of Laser Shock Peening on the Fatigue Behavior of Thin Aluminum Panels, Materials Today: Proceedings, 2015, 2(10)); the peening can be detrimental under some combination of geometry parameters and laser impact energy in thin coupons.

The results show some inconsistencies with other literature: the increase of the laser power density and the number of impacts creates higher surface residual compressive stress, while reducing the depth of the residual compressive stress. Many results confirm the surface behavior, while opposite results are achieved through the thickness. Similarly, the presence of multiple impacts in the paper reduces the depth of the residual compressive stress layer. Can this be confirmed through literature?

The experimental verification in the paper is not clear: LSP experiments were conducted on ceramic samples of unknown dimensions. A better description of the experimental test setup is mandatory. Are the laser parameters described the results of an optimization? Why an overlap of 50% has been chosen?

The depth of the compressive residual stresses is very low. Is this due to the specific material behavior?

Reviewer 2 Report

1. It seems that the resolution of the simulation performed is quite low considering its a multiphysics simulation and that too at the scale of ns, so it must require proper analysis with deep mesh to resolve the nano scale time step and high frequency shockwave.

2. The time-dependent pressure distribution doesnt seem to match with the exponential distribution given by equation (5). Rather it should be a high peak with some damping followed by negative pressure.

3. Very little mathematical relations are given to support the numerical simulation. Provide more mathematical analysis with detailed discussion in the section 2.

4. Simulation results are not in close accordance with the experimental results as depicted by figure 8. Please elaborate more on the reasons of differences in the results.

Round 2

Reviewer 1 Report

all the suggestions from the reviewer have been deeply discussed by the authors. The paper has been definitively improved and is worth of publishing

Reviewer 2 Report

The authors have answered to my queries.